# NEURAL OPERATOR-BASED CURRICULUM LEARNING FOR PHYSICS-INFORMED NEURAL NETWORKS

## ABSTRACT

In this paper, we tackle the critical failure modes of Physics-Informed Neural Networks (PINNs), such as spectral bias and ill-conditioning, which lead to poor convergence on complex PDEs. We identify two key shortcomings in existing curriculum learning methods for PINNs: unreliable knowledge transfer between stages and a reliance on manual, ad-hoc curriculum design. To overcome these limitations, we present Neural Operator-based Curriculum Learning (NOCL), a unified framework that leverages Neural Tangent Kernel (NTK) theory to automate curriculum generation and employs neural operators to enable robust, dynamic knowledge transfer across curriculum stages. By dynamically training the operator and filtering data for PINN initialization, our approach ensures scalable and effective learning across progressively difficult tasks. Experiments verify that our proposed NOCL achieves state-of-the-art performance, markedly improving convergence and generalization over existing methods.

## 1 INTRODUCTION

The advent of neural networks has transformed scientific computing, introducing data-driven methodologies for solving partial differential equations (PDEs). Whereas conventional techniques such as the Finite Element Method (FEM) (Zienkiewicz et al., 2013) and Finite Difference Method (FDM) (LeVeque, 2007) are constrained by their dependence on spatial discretization, which is a major limitation in high-dimensional settings, neural networks present a mesh-free and highly scalable approach (Lagaris et al., 1998; Han et al., 2018). A leading framework in this domain is Physics-Informed Neural Networks (PINNs) (Raissi et al., 2019; Karniadakis et al., 2021), which incorporate physical constraints directly into the training loss. This allows PINNs to address both forward and inverse problems even with sparse data, leading to widespread adoption in disciplines (Mao et al., 2020; Cai et al., 2021; Raissi et al., 2019).

However, training PINNs poses a particularly challenging optimization problem, often suffering from convergence failures in certain parameter regimes (Yan & He, 2024; Krishnapriyan et al., 2021; Chen et al., 2024). To address this, recent studies (Krishnapriyan et al., 2021; Bekele, 2024) have integrated curriculum learning strategies. These include defining a curriculum over the PDE's parameter space to progressively increase complexity, as well as employing causal learning, which structures the problem into sequential temporal stages (Guo et al., 2025; Wang et al., 2024). While these methods can improve convergence, they remain prone to failure or non-convergence in many practical PDE fitting scenarios (Monaco & Apiletti, 2023), highlighting the need for more robust curriculum learning algorithms.

Two fundamental challenges must be overcome to address these limitations effectively (Monaco & Apiletti, 2023). Automatically designing an optimal parameter-based curriculum for a given PDE system is highly non-trivial. Second, PDE solutions are often highly sensitive to parameter changes (Hanna et al., 2024). In Section 4.2, we empirically demonstrate that directly transferring network parameters between different parametric instances results in significant error. This underscores the critical need for a reliable knowledge transfer mechanism across curriculum stages.

In this paper, we propose a novel Neural Operator-based Curriculum Learning framework (NOCL) for training PINNs. Our approach leverages the complementary strengths of Neural Tangent Kernel (NTK) theory for curriculum design and neural operators for robust knowledge transfer.

Our framework is motivated by two key insights:

1. **Neural Operators for Transfer:** Neural operators learn mappings between function spaces, exhibiting strong generalization across PDE instances (Li et al., 2020a; 2023; Lu et al., 2021). This makes them ideal for transferring information between curriculum stages, providing a continuous and informative prior. However, their standard supervised training requires large, pre-computed datasets, which are often unavailable, leading to poorly generalized operators.

2. **NTK for Curriculum Design:** NTK analysis provides a principled way to characterize training dynamics (Tan & Liu, 2024; Jacot et al., 2018). The eigenvalues of the NTK matrix have been shown to correlate with the training difficulty of specific PDE parameters (Wang et al., 2022), offering a foundation for systematic curriculum design. However, the variability of NTK spectra across different PDE systems necessitates a universal metric for quantifying prediction difficulty.

To overcome the data dependency of neural operators, we develop a strategy to dynamically train the operator alongside the curriculum, augmented by a data filtering algorithm. For curriculum design, we introduce a novel, universal metric based on NTK eigenvalue variance to consistently quantify PDE difficulty. Our **key contributions** are summarized as follows:

1. **Identification of Pathological Failure in PINN Curriculum Learning:** We empirically demonstrate that traditional curriculum learning, which retains model parameters across stages, can be ineffective and even detrimental, leading to catastrophic training failure in specific parameter regimes.

2. **Robust Knowledge Transfer via Neural Operators:** We introduce neural operators as a tool for transferring knowledge between curriculum stages. Our method ensures scalability through dynamic operator updates and a data filtering algorithm, effectively mitigating parameter sensitivity issues. This approach is compatible with causal learning frameworks.

3. **Principled Curriculum Design via NTK Analysis:** We establish a principled curriculum design methodology by computing the variance of NTK matrix eigenvalues across PDE parameters. This metric reliably reflects training difficulty, enabling the automatic construction of effective curricula, as validated by our experiments.

## 2 RELATED WORK

### 2.1 PHYSICS-INFORMED NEURAL NETWORKS

Consider a general PDE system defined on a domain $\Omega \subset \mathbb{R}^d$ with boundary $\partial\Omega$:

$$\mathcal{L}[u(\mathbf{x})] = f(\mathbf{x}), \ \mathbf{x} \in \Omega, \quad \text{and} \quad \mathcal{B}[u(\mathbf{x})] = g(\mathbf{x}), \ \mathbf{x} \in \partial\Omega, \tag{1}$$

where $\mathcal{L}$ denotes the differential operator, $\mathcal{B}$ represents boundary operators, $f$ and $g$ are known source terms, and $u : \mathbb{R}^d \to \mathbb{R}$ is the unknown solution function.

PINNs employ a deep neural network $\hat{u}(\mathbf{x}; \boldsymbol{\Theta})$ parameterized by $\boldsymbol{\Theta}$ to approximate the true solution $u(\mathbf{x})$. The network architecture typically consists of multiple fully-connected layers with nonlinear activation functions, enabling the model to capture complex solution patterns across the domain.

The training objective combines multiple physics-informed loss components to ensure the neural network satisfies both the governing equations and boundary constraints:

**Physics Loss:** The physics loss enforces the PDE residual at interior points:

$$\mathcal{L}_{\text{physics}}(\boldsymbol{\Theta}) = \frac{1}{N_p} \sum_{i=1}^{N_p} |\mathcal{L}[\hat{u}(\mathbf{x}_i; \boldsymbol{\Theta})] - f(\mathbf{x}_i)|^2, \tag{2}$$

where $\{\mathbf{x}_i\}_{i=1}^{N_p}$ are collocation points sampled from $\Omega$, and $N_p$ denotes the number of physics points.

**Boundary Loss:** The boundary loss ensures compliance with boundary conditions:

$$\mathcal{L}_{\text{boundary}}(\boldsymbol{\Theta}) = \frac{1}{N_b} \sum_{j=1}^{N_b} |\mathcal{B}[\hat{u}(\mathbf{x}_j; \boldsymbol{\Theta})] - g(\mathbf{x}_j)|^2, \tag{3}$$

where $\{\mathbf{x}_j\}_{j=1}^{N_b}$ are boundary points, and $N_b$ represents the number of boundary points.

The composite loss function combines these components with appropriate weighting:

$$\mathcal{L}_{\text{total}}(\boldsymbol{\Theta}) = \lambda_r \mathcal{L}_{\text{physics}}(\boldsymbol{\Theta}) + \lambda_b \mathcal{L}_{\text{boundary}}(\boldsymbol{\Theta}), \tag{4}$$

where $\lambda_r$ and $\lambda_b$ are hyperparameters that balance the relative importance of physics and boundary constraints. The optimization process minimizes $\mathcal{L}_{\text{total}}(\boldsymbol{\Theta})$ using gradient-based methods.

## 2.2 NEURAL OPERATORS

Neural operators represent a significant advance in deep learning, designed to learn mappings between function spaces rather than finite-dimensional vector spaces. Unlike traditional neural networks that operate on discrete data points, neural operators directly learn the underlying functional relationships, enabling them to generalize to unseen input functions and make predictions at different resolutions without retraining (Li et al., 2020b).

Given a function $u(x)$ defined on domain $V$, the goal of a neural operator $G_\theta$ (parameterized by $\theta$) is to approximate the physical operator $G$ mapping $u(x)$ to another function $v(y)$ defined on domain $K$:

$$G(u) \approx G_\theta : u \mapsto v. \tag{5}$$

Currently, various neural operators have demonstrated significant potential in PDE prediction, such as DeepONet (Lu et al., 2021), Graph Neural Operator (GNO) (Li et al., 2020b), and Fourier Neural Operator (FNO) (Li et al., 2020a). Our motivation for employing a neural operator is to learn the mapping from the parameter space of PDEs to their solution space, thereby enabling effective information transfer across curriculum stages. In subsequent experiments, we select the most commonly used FNO due to its efficiency and strong generalization capabilities.

## 3 METHODOLOGY

PINNs often encounter significant optimization difficulties when solving PDEs with large parameter values, despite showing good performance on simpler problems with smaller parameters (Krishnapriyan et al., 2021). To mitigate issues, curriculum learning has been introduced as a training strategy that promotes gradual learning from easier to more difficult parameter settings (Krishnapriyan et al., 2021). However, its application to PINNs remains challenging due to two major limitations: first, curriculum learning may become ineffective or even detrimental for certain parameter configurations; second, designing consistent and effective curricula for multi-parameter PDE systems, where parameters may interact or conflict in shaping solution difficulty, poses a substantial methodological gap (Monaco & Apiletti, 2023). To address these problems, we introduce a neural operator-based curriculum learning algorithm. This approach employs neural operators as a transfer mechanism between successive curriculum stages and relies on the analysis of the NTK across different PDE parameters to guide the design of the curriculum.

### 3.1 NEURAL OPERATOR-BASED CURRICULUM LEARNING (NOCL)

A standard practice in curriculum learning for PDEs is to initialize the model for a new curriculum stage by directly inheriting the parameters from a model trained on a previous, easier stage. However, our experiments (Section 4.2) reveal that this direct parameter inheritance is often unsuitable for PINNs and can even lead to catastrophic convergence failures. The root cause is that PINNs trained on different PDE parameters learn distinct, often incompatible, solution representations. Directly transferring these parameters results in a poor initialization, causing training instability and suboptimal solutions. To overcome this fundamental limitation, we propose the Neural Operator-based Curriculum Learning (NOCL) algorithm. Instead of transferring model parameters, NOCL employs a neural operator as a robust information transfer tool between curriculum stages, enabling effective knowledge propagation across varying PDE parameter configurations.

The overall procedure of NOCL is illustrated in Figure 1 and detailed in Algorithm 1. It consists of two main phases: an initialization phase and a curriculum loop phase. The initialization phase bootstraps the process by training PINNs directly on a set of initial PDE parameters, typically chosen for their ease of training. After each PINN is trained to a low relative $L_2$ error, we save its

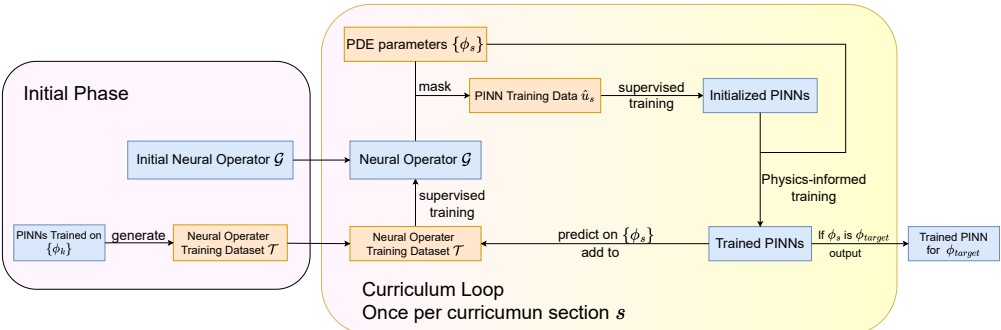

Figure 1: This figure illustrates the Neural Operator-based Curriculum Learning (NOCL). Orange represents datasets, blue represents neural network models. The Initial Phase first trains PINNs on initial PDE parameter subsets $\phi_k$ to generate a training dataset $\mathcal{T}$ for the neural operator $\mathcal{G}$. The Curriculum Loop executes per curriculum section: neural operator $\mathcal{G}$ learns parameter-to-solution mapping via supervised training on $\mathcal{T}$; for PDE parameters $\phi_s$ in this Loop, predicted solutions from $\mathcal{G}$ undergo mask algorithm filtering to select low-residual points as quality initialization for PINN training; after physics-informed training, new solutions augment $\mathcal{T}$ to enhance the neural operator. When processing the target parameter $\phi_{target}$, the algorithm outputs the final trained PINN.

predictions on a predefined grid. Each saved data point forms a pair of PDE parameters and their corresponding solution snapshot. This collection of pairs constitutes the initial training dataset for the neural operator (typically an FNO), which is initialized in this phase.

In the curriculum loop phase, the target PDE parameters are partitioned into a curriculum of progressively more difficult stages. For each stage: First, the neural operator is trained in a supervised manner on its current dataset to learn the mapping from PDE parameters to their solutions; Second, for each new PDE parameter in the current stage, the trained neural operator predicts the corresponding solution function. The resulting $(\mathbf{x}, u(\mathbf{x}))$ pairs are used to initialize the PINN; Third, the PINN continues its training using the standard physics-informed loss defined in (4); Finally, after PINN training is complete, its predictions are evaluated on the grid. This new solution data is added to the neural operator's training set to enhance its generalization for subsequent stages.

Note that neural operator predictions can be inaccurate due to limited training data and resolution dependencies. To ensure only high-quality data initializes the PINN, we propose a mask-based filtering algorithm. This algorithm computes the PDE residual at each collocation point using finite differences. A mask then selects only the top–$\alpha$ fraction of points with the smallest residuals, filtering out regions where predictions deviate significantly from the physics. This step is critical for providing a physically consistent starting point for PINN training.

---

**Algorithm 1** Neural Operator-based CL

**Require:** Curriculum parameters $\{\phi_k\}_{k=1}^{M}$, mask ratio $\alpha$
**Ensure:** Trained PINN for target PDE parameters
1: **Step 1: Initialization Phase**
2: Initialize neural operator $\mathcal{G}$ (typically FNO)
3: Initialize training set $\mathcal{T} = \emptyset$
4: **for** each parameter $\phi_k$ in initialization subset **do**
5:    Train PINN on PDE with parameters $\phi_k$
6:    Generate predictions $u_k(x, t)$ on grid points
7:    Add $(\phi_k, u_k(x, t))$ to $\mathcal{T}$
8: **end for**
9: **Step 2: Curriculum Loop Phase**
10: **for** each curriculum section $s$ **do**
11:    Train neural operator $\mathcal{G}$ on training set $\mathcal{T}$
12:    **for** each parameter $\phi_s$ in section $s$ **do**
13:       Predict function using neural operator: $\hat{u}_s = \mathcal{G}(\phi_s)$
14:       Compute PDE residual error: $R(x) = \mathcal{L}[\hat{u}_s(\mathbf{x})] - f(x)$
15:       Apply mask filtering: retain top $\alpha$ fraction of points with smallest $|R(x, t)|$
16:       Initialize PINN with filtered predictions
17:       Train PINN using PDE loss equation 4
18:       Generate predictions on grid points: $u_s(x, t)$
19:       Add $(\phi_s, u_s(x, t))$ to $\mathcal{T}$
20:    **end for**
21: **end for**
22: **return** Trained PINN for $\phi_{\text{target}}$

---

Our algorithm offers several key advantages over traditional curriculum learning. First, neural operators learn the underlying solution opera-

tor, allowing them to make informed predictions for new parameters, even when the solution space changes significantly. This makes robust knowledge transfer possible. Second, the residual-based filtering prevents the propagation of unphysical errors, mitigating the overfitting and sensitivity issues common in parameter-transfer methods. Third, once trained, the neural operator and its dataset can be reused for new parameter configurations without restarting the entire curriculum, offering a substantial computational advantage over traditional methods that must repeat the process from scratch. As demonstrated in Section 4, NOCL achieves robust performance even with suboptimal curriculum designs, highlighting its practical effectiveness.

## 3.2 NTK-BASED CURRICULUM DESIGN

While NOCL demonstrates robust performance even with suboptimal curricula, a principled method for designing curricula across PDE parameters is essential to prevent initialization failures. To this end, we propose using Neural Tangent Kernel (NTK) eigenvalue analysis for systematic curriculum design. The NTK matrix $K_{\boldsymbol{\theta}}$ for a neural network with parameters $\boldsymbol{\theta}$ is defined as:

$$K_{\boldsymbol{\theta}}(\mathbf{x}, \mathbf{x}') = \langle \nabla_{\boldsymbol{\theta}} f(\mathbf{x}; \boldsymbol{\theta}), \nabla_{\boldsymbol{\theta}} f(\mathbf{x}'; \boldsymbol{\theta}) \rangle, \tag{6}$$

where $f(\mathbf{x}; \boldsymbol{\theta})$ is the neural network output and $\nabla_{\boldsymbol{\theta}} f$ is the parameter gradient. Training difficulties often arise from imbalanced NTK eigenvalues, as high-frequency solution components typically correspond to larger eigenvalues in the NTK spectrum (Wang et al., 2022). While this suggests the NTK condition number could indicate training difficulty, we find it unsuitable for our purpose. Specifically, the condition number is defined as:

$$\kappa(K_{\boldsymbol{\theta}}(\boldsymbol{\phi})) = \frac{\lambda_{\max}(K_{\boldsymbol{\theta}}(\boldsymbol{\phi}))}{\lambda_{\min}(K_{\boldsymbol{\theta}}(\boldsymbol{\phi}))},$$

where $\lambda_i(K_{\boldsymbol{\theta}}(\boldsymbol{\phi}))$'s are the eigenvalues of the NTK matrix for PDE parameters $\boldsymbol{\phi}$. Using the NTK condition number for curriculum design is problematic: minimum eigenvalues often decay to zero with increasing collocation points, rendering the condition number unstable, while relying solely on maximum eigenvalues ignores the spectral imbalance that dictates training difficulty. To overcome these issues, we propose the variance of the NTK eigenvalues as a robust curriculum metric:

$$\mathcal{D}(\boldsymbol{\phi}) = \mathrm{Var}(\lambda_i^{\mathrm{res}}(K_{\boldsymbol{\theta}}(\boldsymbol{\phi}))) + \mathrm{Var}(\lambda_i^{\mathrm{bc}}(K_{\boldsymbol{\theta}}(\boldsymbol{\phi}))),$$

where $\lambda_i^{\mathrm{res}}$ and $\lambda_i^{\mathrm{bc}}$ represent the eigenvalues of the NTK matrix corresponding to the residual and boundary conditions, respectively. The rationale for this design is that the NTK spectrum is known to decay rapidly (Wang et al., 2022). In such a distribution, the maximum eigenvalue, acting as an outlier due to its large magnitude, naturally contributes a dominant term to the variance calculation because variance scales with the square of the distance from the mean. Furthermore, for a given maximum eigenvalue, a faster decay rate, which indicates greater spectral imbalance, results in a larger variance. Consequently, the sum of variances yields a robust measure of spectral imbalance that remains stable even as minimal eigenvalues decay, effectively capturing the uneven distribution that characterizes training difficulty.

Our curriculum design procedure is as follows. We begin by constructing a candidate set of curricula for the relevant PDE parameters (e.g., residual parameters and initial/boundary condition parameters). This is typically done by sampling parameter values along a path from zero to the target value $\boldsymbol{\phi}_{\mathrm{target}}$. For each sampled parameter value $\boldsymbol{\phi}_i$, we compute the corresponding NTK eigenvalue variance $\mathcal{D}(\boldsymbol{\phi}_i)$. We then filter this set, excluding any samples where the variance exceeds the target's variance, $\mathcal{D}(\boldsymbol{\phi}_{\mathrm{target}})$. This ensures the curriculum consists only of scenarios that are simpler than or equally complex to the target. Subsequently, we analyze the variance trends across the filtered candidate parameters. The parameter exhibiting the largest incremental change in variance is selected as the primary curriculum parameter. The curriculum for this parameter is then defined by the sequence of values where $\mathcal{D}(\boldsymbol{\phi}) \leq \mathcal{D}(\boldsymbol{\phi}_{\mathrm{target}})$, guaranteeing a progressive increase in difficulty. A key advantage of this NTK-based analysis is its ability to design effective curricula even when training difficulty does not increase monotonically with the parameter. In such cases of non-monotonic increase, the initial stage of the curriculum is not blindly set to a value near zero. Instead, we select the parameter region with the smallest variance, $\min \mathcal{D}(\boldsymbol{\phi})$, as the starting point, ensuring the learning process begins with the genuinely easiest instance.

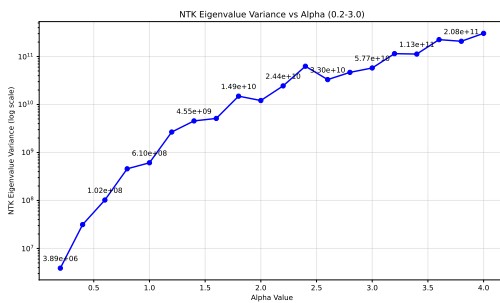 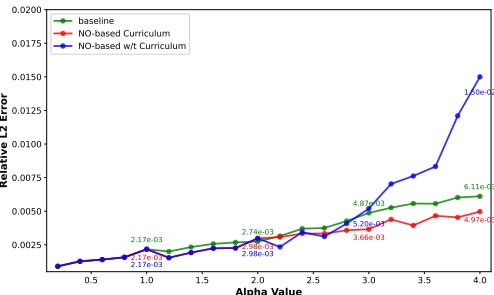

(a) NTK eigenvalue variance analysis for the 3D heat equation across different alpha values. The plot shows how the variance of NTK eigenvalues changes as the heat conduction parameter $\alpha$ varies from 0.2 to 4.0.

(b) Comparison of relative L2 errors for different methods on the 3D heat equation across varying alpha values. The plot compares three approaches: baseline PINN (green), NO-based curriculum learning (red), and NO-based without curriculum learning (blue).

Figure 2: Analysis results for the 3D heat equation: (a) NTK eigenvalue variance analysis across different $\alpha$ values, and (b) method comparison of relative $L_2$ errors across different $\alpha$ values.

## 4 EXPERIMENTS ON PDEs

### 4.1 HEAT EQUATION

To demonstrate the effectiveness of our NOCL, we perform numerical experiments on the two-dimensional heat equation, which is a fundamental model for thermal and general diffusion processes. The governing PDE is:

$$\frac{\partial u}{\partial t} - \alpha \left( \frac{\partial^2 u}{\partial x^2} + \frac{\partial^2 u}{\partial y^2} \right) = 0, \quad x, y \in [0, 1], \ t \in [0, 0.1] \tag{7}$$

subject to the initial condition $u(x, y, 0) = \sin(\pi x)\sin(\pi y)$ and Dirichlet boundary conditions $u(0, y, t) = u(1, y, t) = u(x, 0, t) = u(x, 1, t) = 0$. The thermal diffusivity is denoted by $\alpha$. The analytical solution is $u^*(x, y, t) = \sin(\pi x)\sin(\pi y)\exp(-2\alpha\pi^2 t)$.

We begin by substantiating the utility of the NTK eigenvalue variance (Section 3.2) as a principled indicator of training difficulty. Although the heat equation involves only a single thermal coefficient $\alpha$—so NTK eigenvalue variance–based parameter selection is not strictly required—we include this analysis as a sanity check. By evaluating NTK variance over $\alpha \in [0.2, 4.0]$, we confirm that its trend aligns closely with observed training error, thereby validating our proposed curriculum ordering. This experiment further shows that NOCL improves convergence even in this simple single-parameter setting. For 20 uniformly spaced values, we compute the NTK eigenvalue variance using the experimental setup described in Appendix A.1.

As shown in Figure 2a, the variance displays a variation pattern over $\alpha$ that is similar to that of the final relative $L_2$ error (green curve in Figure 2b). This strong correlation supports NTK variance as an effective proxy for training difficulty. To rule out the confounding effect of uniform spectral scaling, which can also inflate variance, we further compute the variance of the logarithm of the eigenvalues, restricting to the top $75\%$ to mitigate numerical instability from near-zero values. The resulting log-variance (Figure 4, Appendix A.2) aligns with both the standard variance and the error trend in Figure 2b, indicating that NTK variance captures a meaningful notion of spectral imbalance suitable for curriculum construction.

We then evaluate our NOCL over the same range of $\alpha$, holding hyperparameters and training configurations fixed (Appendix A.1). As shown in Figure 2b, NOCL consistently attains lower relative $L_2$ error than the baseline for all $\alpha$ except $\alpha = 2$. The improvements are modest for smaller $\alpha$ and become increasingly pronounced as $\alpha$ grows, indicating that the proposed curriculum effectively mitigates failure modes of the base model in more challenging PDE parameter regimes.

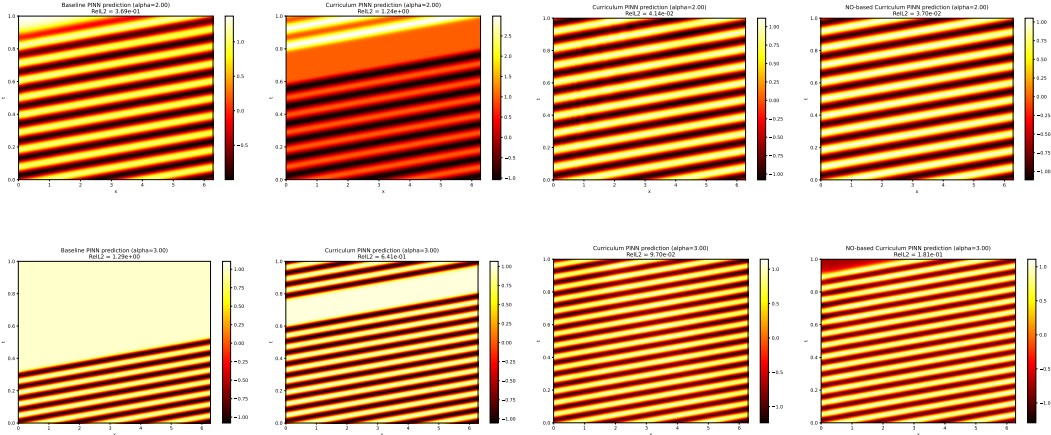

Figure 3: Visualization results for convection equation with varying initial condition parameter $\alpha$. The first row shows results for $\alpha = 2$, while the second row shows results for $\alpha = 3$. Within each row, from left to right: baseline PINN predictions, traditional curriculum learning PINN predictions, NOCL PINN predictions, and NOCL without mask algorithm PINN predictions.

To validate the necessity of curricular training for the neural operator within the NOCL framework, we conduct an ablation study in which the operator is trained solely on the dataset generated in the initial stage (i.e., data corresponding to $\alpha \leq 1$), without any further data expansion or neural operator updates during the subsequent curriculum stages. Experimental results show that performance degrades severely on larger $\alpha$ values (blue curve in Figure 2b). This outcome indicates that training the neural operator only on a limited parameter range leads to insufficient generalization capability due to inadequate data coverage, thereby confirming the necessity of progressively expanding the training set and continuously updating the neural operator throughout the curriculum stages.

## 4.2 Convection Equation

To demonstrate the superiority of our NOCL compared to traditional CL, we conduct experiments on the convection equation, which is fundamental for modeling transport phenomena. We employ the convection equation system from the study by Krishnapriyan et al. (2021). The convection equation is defined as:

$$\frac{\partial u}{\partial t} + \beta \frac{\partial u}{\partial x} = 0, \quad x \in [0, 2\pi], \quad t \in [0, 1] \tag{8}$$

with initial condition $u(x, 0) = \sin(x)$ and periodic boundary conditions, where $\beta$ is the convection coefficient.

We use a 4-layer MLP with 64 hidden units as our base model architecture. In NOCL, since the convection equation system contains only a single parameter $\beta$, we only need to perform the second step of NTK analysis. Based on the NTK matrix generated from 64 data points, we observe that the eigenvalue variance of the NTK matrix increases monotonically with $\beta$. We design the corresponding curriculum based on this result, with detailed curriculum specifications provided in Appendix B.1.

For comparison, we implement traditional curriculum learning settings that are identical to NOCL, except for the neural operator-related components. The baseline settings use the same physics training configuration as our approach.

The experimental results are presented in Table 1. In the curriculum learning experiments, NOCL achieves the lowest relative $L_2$ errors for $\beta$ values of 40, 50, 60, and 70. Notably, at $\beta = 40$, our method provides only modest improvements, suggesting that the problem difficulty at this parameter value is within the capability of traditional approaches. However, for more challenging problems with $\beta$ values of 50, 60, and 70, our method demonstrates significant improvements, indicating strong scalability of our approach on difficult curriculum stages.

Table 1: Relative $L_2$ errors for convection equation with varying $\beta$ parameter. Methods: Baseline, curriculum learning without neural operator (CL w/o NO), proposed NOCL algorithm, and its variant without data filtering (w/o mask).

| Method | $\beta = 40$ | $\beta = 50$ | $\beta = 60$ | $\beta = 70$ |
|---|---|---|---|---|
| Baseline | 1.52e-2 | 2.60e-2 | 2.91e-2 | 6.33e-1 |
| CL w/o NO | 1.01e-2 | 1.32e-2 | 2.18e-2 | 2.82e-2 |
| NOCL | **8.61e-3** | **5.98e-3** | 1.61e-2 | **1.24e-2** |
| w/o mask | 9.71e-3 | 7.40e-3 | **1.22e-2** | 1.57e-2 |

Table 2: Relative $L_2$ errors for convection equation with varying initial condition parameter $\alpha$. When $\alpha$ equals 3, the results obtained without using the data filtering algorithm exhibit large errors.

| Method | $\alpha = 2$ | $\alpha = 3$ |
|---|---|---|
| Baseline | 3.69e-1 | 1.29e+0 |
| CL w/o NO | 1.24e+0 | 6.41e-1 |
| NOCL | 4.14e-2 | **9.07e-2** |
| w/o mask | **3.70e-2** | 1.81e-1 |

Table 3: Relative $L_2$ errors for reaction-diffusion equation with different curriculum parameters and methods. "MLP" denotes using MLP as the base model, while "Causal" refers to using MLP with causal training as the base model for curriculum learning. "Baseline" refers to the approach that does not employ any curriculum learning methods.

| Method | Baseline | CL w/o NO | | NOCL | |
|---|---|---|---|---|---|
| | | $\rho$ | $\nu$ | $\rho$ | $\nu$ |
| MLP | 4.78e-1 | 9.93e-2 | 7.92e-2 | **7.36e-2** | 8.10e-2 |
| Causal | 1.00e-1 | 1.16e-1 | 1.00e-1 | 7.86e-2 | **7.60e-2** |

To further validate the superiority of our NOCL algorithm and investigate the pathological issues in traditional curriculum learning for PINNs, we design an additional experiment using the convection system with $\beta$ fixed at 30 and the initial condition modified to $u(x, 0) = \sin(\alpha x)$. In this experiment, we construct a curriculum over the parameter $\alpha$, with detailed specifications provided in Appendix B.1, while maintaining the same hyperparameters as in the $\beta$ experiments. To ensure that the observed performance improvement stems from addressing fundamental PINN training issues rather than merely increased computational cost, we train a baseline model for 500,000 steps, which is ten times longer than PINN training in NOCL.

The experimental results, presented in Figure 3 and Table 2, demonstrate that traditional curriculum learning struggles with this parameter modification. Specifically, at $\alpha = 2$, the neural network produces significant errors in the upper-left region. We attribute this limitation to the analytical solution's heightened sensitivity to changes in $\alpha$ compared to $\beta$. Even with carefully designed small curriculum intervals for $\alpha$, the training at one stage adversely affects subsequent stages. As shown in Figure 5 in Appendix B.2, the extended baseline training (500,000 steps) still fails to converge for $\alpha = 2, 3$, confirming that simply increasing computational budget cannot resolve these pathological training issues. In contrast, NOCL, which utilizes neural operators as information transfer tools combined with mask filtering, effectively prevents the propagation of harmful information, enabling robust and stable curriculum learning across the parameter domain.

We also conduct an ablation study on the mask algorithm in this experiment. From Tables 1, 2, and Figure 3, we observe that for relatively simple PDE systems, using a mask with a retention ratio of 0.5 does not provide significant improvements and may even lead to increased relative L2 errors. However, from the visualization results without the mask algorithm at $\alpha = 3$ in Figure 3, we can observe that PINN produces severe errors in the upper-left region. When PDEs exhibit high-frequency solutions that make it difficult for neural operators to predict relatively accurate values, the mask algorithm can prevent low-quality data from contaminating PINN initialization. Simultaneously, for PDEs with low-frequency solution coefficients, we can select larger retention ratios in the mask algorithm to enhance the effectiveness of NOCL.

### 4.3 REACTION-DIFFUSION EQUATION EXPERIMENT

To further demonstrate the versatility and effectiveness of NOCL, we conduct experiments on the reaction-diffusion equation. We employ the reaction-diffusion equation system from the study by Krishnapriyan et al. (2021), which investigates possible failure modes in PINNs. The reaction-

diffusion equation is defined as follows:

$$\frac{\partial u}{\partial t} - \nu \frac{\partial^2 u}{\partial x^2} - \rho u(1-u) = 0, \quad x \in [0,1], \quad t \in [0,1] \tag{9}$$

with initial condition $u(x,0) = \sin(\pi x)$, where $\nu$ is the diffusion coefficient and $\rho$ is the reaction coefficient. This equation combines diffusion processes (controlled by $\nu$) with nonlinear reaction terms (controlled by $\rho$), creating complex dynamics that are particularly challenging for neural network approximation due to the interplay between these two competing mechanisms.

For this problem, we test three approaches: baseline without curriculum learning, traditional curriculum learning with direct parameter transfer, and NOCL. To evaluate the scalability of curriculum learning, we implement these approaches on both standard MLP models and MLP models trained using causal methods (Wang et al., 2024). The detailed hyperparameters used in our experiments can be found in Appendix C.2.

We test Baseline, traditional curriculum learning and NOCL on a challenging PDE system with parameters $\nu = 4$ and $\rho = 5$. Through analysis of NTK matrix eigenvalues variance, as shown in Figure 6 in Appendix C.1, we discover distinct curriculum stages in the $\nu$ parameter, while no clear curriculum characteristics exist in the $\rho$ parameter. We select $\nu$ as the curriculum parameter for our primary experiments. To compare curriculum strategies designed on different parameters, we also conduct curriculum learning experiments on the parameter $\rho$ for both the traditional method and our proposed NOCL.

The experimental results are presented in Table 3. We observe that regardless of whether MLP or causal training is used as the base model, NOCL achieves the lowest relative $L_2$ errors on this problem. An unexpected result emerges: when using NOCL, the relative $L_2$ errors for curricula designed on $\rho$ are not significantly lower than those for curricula on $\nu$. However, when using traditional curriculum learning methods, curricula on $\nu$ significantly outperform those on $\rho$.

To compare the differences between the two curriculum designs, we examine the relative $L_2$ error of the PINN trained during the initial stage for curricula structured on parameters $\nu$ and $\rho$. We observe that in the initial stage of the curriculum on $\rho$ (with $\rho = 1$, $\nu = 4$), the relative $L_2$ error reaches $1.84 \times 10^{-1}$, indicating that the PINN fails to provide high-quality training data for the neural operator at initial stage. In contrast, for the curriculum on $\nu$ at its initial stage (with $\nu = 1$, $\rho = 5$), the relative $L_2$ error is only $5.97 \times 10^{-2}$. This significant performance gap explains why traditional curriculum learning achieves a lower final relative $L_2$ error when the curriculum is designed on $\nu$, thereby supporting the effectiveness of our proposed NTK eigenvalue variance metric. Regarding the fact that NOCL achieves a similarly low error even when the curriculum is designed on $\rho$, we hypothesize that this is due to the excellent generalization capability of the neural operator and the use of mask algorithms to filter out harmful information.

Furthermore, NOCL demonstrates superior scalability compared to traditional curriculum learning. When using causal learning, traditional curriculum learning algorithms exhibit increased relative $L_2$ errors, while NOCL continues to show decreased relative $L_2$ errors. Therefore, we conclude that NOCL combined with NTK analysis exhibits excellent scalability across multi-parameter PDEs and various base models.

## 5 CONCLUSION

Curriculum learning has demonstrated potential in addressing PDE problems that are challenging to train PINNs. However, existing curriculum learning approaches still face two significant unresolved issues: traditional methods based on direct parameter transfer may fail on certain curricula, and designing curricula for multi-parameter PDEs requires prior knowledge to define an appropriate progression. To tackle these problems, we propose a neural operator-based curriculum learning framework. By dynamically training a neural operator as a transfer tool between curriculum stages and incorporating a data filtering algorithm, our approach ensures effective knowledge propagation across successive stages. Meanwhile, the variance of the eigenvalues of the NTK matrix is introduced to quantify the rationality of the curriculum design. Our results demonstrate that NOCL substantially mitigates key failure modes of traditional curriculum learning across multiple PDE systems, while also validating the effectiveness of NTK-based analysis in curriculum construction.

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

# A HEAT EQUATION

## A.1 HYPER-PARAMETER

For the heat equation, we use a 4-layer MLP with a width of 64 as the base model. We use the Adam optimizer and train for 50,000 steps with an initial learning rate of $1 \times 10^{-3}$, applying an exponential decay of 0.95 every 500 steps. Each batch samples 512 residual points and 250 initial/boundary points. For curriculum learning, we divide the range 0.2 to 4.0 into 20 curriculum levels at intervals of 0.2. The range 0.2 to 1.0 is used as the initial curriculum, and every five curriculum levels thereafter form one section. The neural operator is an FNO with 8 layers and 16 modes. We use the Adam optimizer with an initial learning rate of $1 \times 10^{-3}$, training 3,000 steps between sections and applying an exponential decay of 0.95 every 30 steps. For FNO pretraining to initialize the MLP, we also use Adam with an initial learning rate of $1 \times 10^{-3}$, for a total of 10,000 steps, with an exponential decay of 0.9 every 2,500 steps. We use 0.3 as the keep ratio for the masking algorithm.

## A.2 LOG-EIGENVALUE CURVE

To show that the increased variance is not due to a uniform scaling of eigenvalues, we also computed the variance of the eigenvalues in the log domain. Due to numerical precision and outliers, we retained only the largest 75% of the eigenvalues.

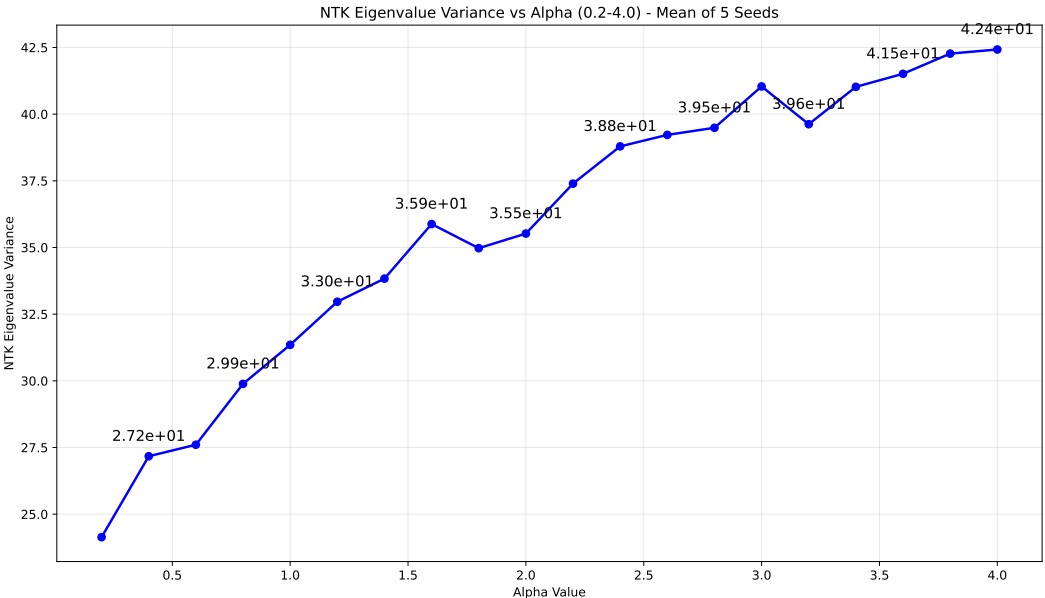

Figure 4: NTK log-eigenvalue variance analysis for the 3D heat equation across different alpha values. The plot shows how the variance of NTK log-eigenvalues changes as the heat conduction parameter $\alpha$ varies from 0.2 to 4.0

# B CONVECTION EQUATION

## B.1 HYPER-PARAMETERS

In the curriculum design for the convection coefficient beta, the initial curriculum is set as linspace(1,30,15), with four subsequent curriculum sections: $(30, 40]$, $(40, 50]$, $(50, 60]$, and $(60, 70]$. Within each section, 5 PINNs are trained using beta values spanning the range (e.g., 32, 34, 36, 38, 40 for the 30-40 section). For the alpha curriculum design, the initial curriculum is set as linspace(0.1,1,15), followed by two sections: $(1, 2]$ and $(2, 3]$, with 5 PINNs trained in each section using corresponding alpha values. The selection of alpha and beta values remains independent of whether the neural operator is employed.

For each PINN training, we use 50,000 physics training steps with the Adam optimizer, initial learning rate of $1 \times 10^{-3}$, and exponential decay of 0.95 every 500 steps. The batch size consists of 256 residual loss data points and 100 initial condition data points. For the neural operator, we employ an 8-layer Fourier Neural Operator with 16 modes. The FNO training details are as follows: we train the FNO for 3,000 epochs using the Adam optimizer with an initial learning rate of $1 \times 10^{-3}$, exponential decay of 0.95 every 100 steps. When using FNO predictions to train PINN networks, the FNO predicts on a $128 \times 128$ grid, and we apply mask filtering with a ratio of 0.5. The PINN initialization uses the Adam optimizer with an initial learning rate of $1 \times 10^{-3}$, 10,000 training steps, and exponential decay of 0.9 every 2,500 steps. We use 0.5 as the keep ratio for the masking algorithm.

## B.2 VISUAL RESULTS FROM EXTENDED TRAINING

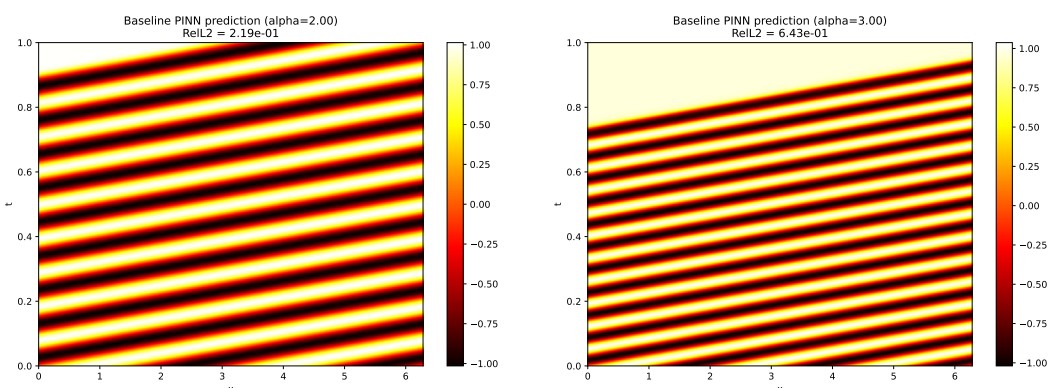

Figure 5: Prediction visualization using the baseline MLP after 500,000 training steps, with results for $\alpha = 2$ (left) and $\alpha = 3$ (right).

## C REACT-DIFFUSION EQUATION

### C.1 NTK ANALYSIS COMPARISON

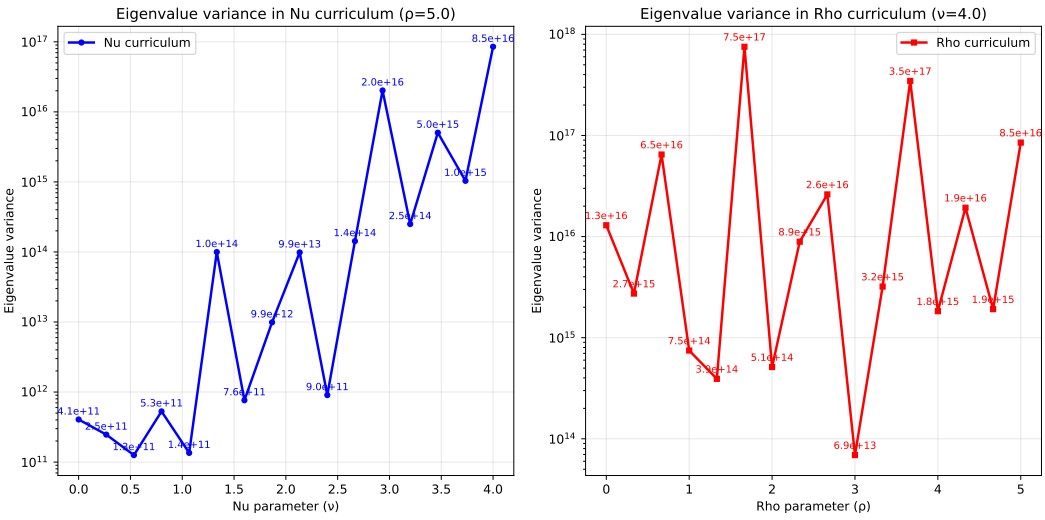

Figure 6: Eigenvalue variance of the NTK across parameter curricula. Left: varying $\nu$ with $\rho$ fixed at 5.0. Right: varying $\rho$ with $\nu$ fixed at 4.0. Curves show the mean variance over 5 random seeds (128 domain points, 32 initial-condition points); values are annotated at each marker, and the y-axis uses a logarithmic scale.

## C.2 HYPER-PARAMETERS

Based on the variance trends observed in Figure 6, we designed the curriculum sections for the parameter $\nu$. The initial section for $\nu$ was set as linspace(0, 1.0, 10), followed by subsequent curriculum sections covering the ranges $(1, 2]$, $(2, 3]$, and $(3, 4]$. Within each section, five distinct values of $\nu$ were selected for PINN training. Similarly, for the parameter $\rho$, the initial section was designed over the range $[0, 1]$ using linspace(0, 1.0, 10), with subsequent sections covering $(1, 2]$, $(2, 3]$, $(3, 4]$, and $(4, 5]$. In each of these sections, five values of $\rho$ were used for PINN training.

For baseline MLP training, we use 50,000 physics training steps with the Adam optimizer, initial learning rate of $1 \times 10^{-3}$, and exponential decay of 0.95 every 500 steps. The batch size consists of 256 residual loss data points and 100 initial condition data points. For the neural operator, we employ an 8-layer Fourier Neural Operator with 16 modes. The FNO training details are as follows: we train the FNO for 3,000 epochs using the Adam optimizer with an initial learning rate of $1 \times 10^{-3}$, exponential decay of 0.95 every 100 steps. When using FNO predictions to train PINN networks, the FNO predicts on a $128 \times 128$ grid, and we apply mask filtering with a ratio of 0.5. The PINN initialization uses the Adam optimizer with an initial learning rate of $1 \times 10^{-3}$, 10,000 training steps, and exponential decay of 0.9 every 2,500 steps. We use 0.5 as the keep ratio for the masking algorithm.

When using causal learning as the base model, we divided the time domain $[0, 1]$ into 5 time windows. The total training steps were set to 50,000, with an additional time window added every 10,000 steps. When the i-th time window was introduced, the learning rate was reset to $(1e - 3) * 0.9^{(i-1)}$.

## D LLM USAGE

In the preparation of this manuscript, the authors employed Claude-4 and DeepSeek-V3.1 for language editing and readability enhancement. The authors conducted thorough review and editing of all AI-assisted content and assume complete responsibility for the publication's content and accuracy.

