# OpenReview forum: "Neural Operator-based Curriculum Learning for Physics-Informed Neural Networks"
_ICLR.cc/2026/Conference — ICLR 2026 Conference Withdrawn Submission_

### Official Review · Reviewer_DWSU · 2025-10-31

**Soundness:** 3
**Presentation:** 3
**Contribution:** 3
**Rating:** 4
**Confidence:** 3

**Summary:**

This paper proposes Neural Operator-based Curriculum Learning (NOCL) to address critical failure modes in Physics-Informed Neural Networks (PINNs) when solving complex PDEs. The authors identify that traditional curriculum learning for PINNs suffers from two key issues: direct parameter transfer between curriculum stages can cause catastrophic failures, and manual curriculum design for multi-parameter systems lacks principled guidance. NOCL addresses these by using neural operators (FNO) to provide robust knowledge transfer across curriculum stages, combined with NTK eigenvalue variance as an automated difficulty metric for curriculum construction. A mask-based filtering algorithm ensures only high-quality predictions initialize PINN training. Experiments on heat, convection, and reaction-diffusion equations demonstrate that NOCL achieves state-of-the-art performance, successfully training PINNs on challenging parameters where both baseline and traditional curriculum learning fail, while showing robustness even with suboptimal curriculum designs.

**Strengths:**

- The paper provides clear empirical evidence that traditional parameter-transfer-based curriculum learning can catastrophically fail, sometimes performing worse than baseline, which motivates the need for alternative approaches.
- Employing neural operators for knowledge transfer between curriculum stages, rather than directly inheriting network parameters, is an innovative solution that effectively addresses the parameter sensitivity problem while maintaining scalability through dynamic operator updates.
- Using NTK eigenvalue variance as a difficulty metric provides a principled and automated approach to curriculum construction based on established NTK theory, eliminating ad-hoc manual design and demonstrating strong empirical correlation with actual training difficulty across multiple PDE systems.

**Weaknesses:**

- The assumption needs to be better justified.
- The experiments need to be elaborated.
- Systematic analysis should be added to the mask filtering.
- The term "knowledge transfer" is misleading.

**Questions:**

- The paper's effectiveness relies on a critical unstated assumption: that initializing PINNs with approximate solution data (x, u(x)) enables convergence on otherwise intractable parameters. This assumption is neither explicitly acknowledged nor independently validated—the paper provides no theoretical analysis, prior literature support, or dedicated experiments isolating this effect. Without establishing when and why such initialization works, the method's applicability and failure modes remain unclear.
- Limited experimental scope with only low-dimensional (1D-2D) problems and relatively simple PDEs. The scalability to high-dimensional systems, industrial-scale problems, or more complex PDE types remains unclear, and computational cost analysis is largely absent.
- The mask filtering algorithm lacks systematic analysis and presents a critical trade-off that is not adequately explored: retaining too few points may provide insufficient initialization data for PINN training, while retaining too many points risks contaminating the initialization with low-quality predictions. The choice of retention ratio (α=0.5) appears arbitrary with no sensitivity analysis across different values, and the conditions determining when masking is necessary versus harmful are not clearly characterized, making it difficult to apply the method to new problems.
- The paper misleadingly frames the approach as "knowledge transfer" (abstract, Section 3.1). The method does not transfer learned representations or model parameters; instead, neural operators provide numerical solution predictions (x, u(x)) pairs to initialize fresh PINNs that are then trained from scratch. This is fundamentally warm-start optimization rather than transfer learning, as each PINN learns independently without inheriting structural knowledge from previous stages. The approach would be more accurately described as "solution-based initialization."

---

### Official Review · Reviewer_wxCg · 2025-11-01

**Soundness:** 2
**Presentation:** 2
**Contribution:** 2
**Rating:** 4
**Confidence:** 5

**Summary:**

Existing curriculum learning approaches in PINN still face two significant unresolved issues: traditional methods based on direct parameter transfer may fail on certain curricula, and designing curricula for multi-parameter PDEs requires prior knowledge to define an appropriate progression. This paper proposes a neural operator-based curriculum learning framework, which could ensure effective knowledge propagation across successive stages.

**Strengths:**

1. This paper integrates neural operators with curriculum learning to overcome the knowledge transfer problem between curriculum stages.
2. The motivation of the paper is valid.
3. The overall scheme of the manuscript is reasonable.

**Weaknesses:**

1. The paper compares primarily against baseline PINNs and traditional curriculum learning, but fails to compare with recent advanced PINN variants.
2. The paper claims to solve two critical issues with existing curriculum learning methods (unreliable knowledge transfer and manual design), but the experimental evidence for these specific contributions is weak. For instance, the ablation study on mask filtering (Table 1 vs. Table 2) suggests that the mask algorithm might be more critical than the neural operator itself, contradicting the paper's emphasis on neural operators for knowledge transfer.
3. The experimental comparisons were insufficient, lacking more quantitative indicators for comparison.
4. The results in Table 2 and Figure 3 show that simply adding the mask algorithm (w/o NO) already provides significant improvements over traditional curriculum learning, suggesting that the neural operator component may not be as essential as claimed.
5. The author lacks a comparison with existing curriculum  learning on the PINN method, such as CoPINN: Cognitive Physics-Informed Neural Networks.

**Questions:**

See Weaknesses.

---

### Official Review · Reviewer_wW82 · 2025-11-05

**Soundness:** 2
**Presentation:** 2
**Contribution:** 2
**Rating:** 2
**Confidence:** 3

**Summary:**

This paper proposes a neural operator-based curriculum learning framework (NOCL) to improve the generalization ability of Physics-Informed Neural Networks (PINNs) under multi-parameter Partial Differential Equations (PDEs). The key idea is to first train a neural operator to learn a parameter-to-solution mapping, and then use its predictions—filtered by residual-based masking—to initialize subsequent PINN training stages. The paper further introduces a task difficulty metric based on the variance of the Neural Tangent Kernel (NTK) eigenvalue spectrum to determine curriculum ordering. Experiments show improved performance on selected benchmark tasks.

**Strengths:**

1.The paper explores the integration of neural operators and curriculum learning to address PINN generalization in multi-parameter PDE problems. The proposed workflow is interesting and shows some empirical improvement.

2.The motivation is clearly presented. It targets well-known failure modes in PINNs such as spectral bias, parameter sensitivity, and optimization ill-conditioning, and proposes corresponding mitigation techniques.

3.The introduction of a quantifiable curriculum metric using NTK eigenvalue variance instead of heuristic ordering is insightful.

**Weaknesses:**

1.The reliability of residual masking hinges on the assumption of solution structural consistency across stages. If this assumption fails, the method may degrade or even collapse. Even if the solution shape remains similar, the new model’s optimization dynamics may differ, and previous “good” points might no longer be effective. This assumption demands stronger theoretical or structural support, which the paper currently does not provide. Furthermore, masking is applied to the output of a black-box neural operator (FNO) that does not inherently enforce physical consistency. A low residual may occur by chance rather than indicating physical correctness. Thus, the core curriculum strategy lacks robustness and generality.

2.While PINN training is physically constrained, the transfer mechanism itself lacks physical constraints. Therefore, the success of the transfer depends heavily on the neural operator's ability to capture solution structure. The use of “low-residual points” from a previous stage’s neural operator prediction as initialization for the next stage is not guaranteed to be reliable—especially under significant parameter shifts. Theoretically, this residual-masking transfer mechanism is unstable and lacks justification. Without theoretical or statistical analysis on cross-stage consistency, such point-wise knowledge transfer remains empirical and fragile. Moreover, the demonstrated success occurs only under controlled conditions. This does not generalize to broader settings with abrupt structural changes in the solution space. The lack of theoretical support for cross-structure transferability is a major limitation.

3.The paper lacks sufficient ablation studies to validate the method. What happens if NTK-based ordering is removed? Does residual masking alone help? Would uniform initialization perform similarly? These questions remain unanswered.

4.Although the authors claim to perform function-level transfer, in essence, they still rely on a trained model (the neural operator), which is itself a parametric neural network. Therefore, this is not truly model-free transfer.

5.The paper introduces a large number of complex-sounding terms (e.g., “function-level transfer,” “spectral bias avoidance,” “residual masking initialization,” “neural-operator curriculum learning”), which may obscure the true contribution. A more concise presentation focused on core ideas is recommended.

6.Typographical error in Figure 1: “curricumun” should be corrected to “curriculum.” The figure layout is cluttered and deserves a cleaner redesign.

**Questions:**

1.Can the residual masking mechanism be theoretically justified? Is there any statistical or functional consistency analysis available across different PDE parameters?

2.Figure 6 is meant to support curriculum ordering, but it seems to suggest that NTK spectral structures vary significantly across parameters. Doesn’t this contradict the assumption of mask consistency?

3.NTK computation is extremely expensive in large-scale models or datasets. How does the proposed method scale in such cases? Are there more efficient ways to perform difficulty-aware curriculum planning?

---

### Official Review · Reviewer_yC3g · 2025-11-11

**Soundness:** 3
**Presentation:** 3
**Contribution:** 3
**Rating:** 6
**Confidence:** 3

**Summary:**

This paper addresses two core failure modes of Physics-Informed Neural Networks (PINNs) on complex PDEs — spectral bias and ill-conditioning — which often cause poor convergence. It identifies two major limitations of existing curriculum learning (CL) approaches for PINNs: (1) unreliable knowledge transfer between curriculum stages and (2) manual, ad-hoc curriculum design.
To overcome these issues, the authors propose Neural Operator-based Curriculum Learning (NOCL), a unified framework that leverages neural operators (mainly FNO) to perform functional-space knowledge transfer between stages, and uses the variance of the Neural Tangent Kernel (NTK) spectrum as a universal difficulty measure for automated curriculum construction. Additionally, PDE residual–based masking is used to filter high-quality points for PINN initialization. Experiments on heat, convection, and reaction–diffusion equations show that NOCL significantly improves convergence and generalization compared to baselines.

**Strengths:**

1. **Clear motivation and problem definition.**
   The paper systematically analyzes the two weaknesses of curriculum learning (CL) in PINNs — parameter inheritance instability and manual curriculum tuning — and provides empirical evidence supporting these claims.

2. **Sound method design.**
   - Neural operators enable function-space transfer, avoiding parameter compatibility issues.
   - The PDE-residual-based mask filters unreliable operator outputs during initialization.
   - The NTK variance serves as a task-agnostic and theoretically interpretable difficulty metric, improving curriculum robustness.

3. **Coherent algorithmic structure.**
   Algorithm 1 and Figure 1 clearly describe the closed-loop workflow (*train operator → operator-guided initialization → physics training → feedback*).
   This makes the framework reproducible and extensible.

4. **Comprehensive experiments.**
   - **Heat equation:** NTK variance correlates with training error; NOCL yields consistent L2 improvements.
   - **Convection equation:** Better performance under large β and multi-stage α curricula, with masking ablation.
   - **Reaction–diffusion:** Strong results under both ν and ρ curricula, outperforming “causal” CL.

5. **Rich implementation details.**
   The appendix includes hyperparameters, curriculum segmentation, and sampling strategy, which facilitate reproduction and adaptation to other PDE problems.

**Weaknesses:**

1. **Limited novelty boundary and comparison depth.**
   While the combination of operator + curriculum + mask + NTK variance is well-motivated, the paper lacks direct comparisons to *operator-enhanced PINNs* (e.g., PINO, DeepONet with physics constraints) or *NTK/spectral-based curriculum methods*.
   Claims of SOTA are only relative to weaker baselines, leaving uncertainty about the position of NOCL among stronger contemporaries.

2. **Weak theoretical justification of NTK variance.**
   The metric’s robustness under varying network width, sampling density, or scaling is not systematically analyzed.
   Evidence remains empirical, without formal generalization guarantees or complexity estimates for large-scale PDE systems.

3. **Potential bias in operator training.**
   The initial operator is trained on PINN predictions rather than high-fidelity ground truth, introducing a “bootstrapping” bias.
   Although masking mitigates this, quantifying or comparing against hybrid (small true data + bootstrap) strategies would strengthen the paper’s reliability.

4. **Lack of efficiency evaluation.**
   The experiments only report relative L2 errors.
   There are no metrics on wall-clock time, GPU hours, convergence speed, or curriculum length–performance trade-offs, which are crucial for evaluating the practicality of NOCL.

5. **Incomplete ablations.**
   - *CL w/o NO* only removes the operator component; more variants (e.g., using different difficulty metrics like curvature or loss spectral energy) should be tested.
   - The mask ratio α (0.3/0.5) is fixed; a sensitivity analysis or robustness curve is missing, which limits understanding of hyperparameter stability.

**Questions:**

1. **On the NTK variance metric:**
   Does its correlation with task difficulty hold under different network widths, activations, or sampling densities?
   Have you validated its consistency across multiple architectures or tasks?

2. **Generalization to complex PDEs:**
   How does NOCL behave on highly nonlinear, multi-scale, or multi-physics PDEs (e.g., Navier–Stokes, Allen–Cahn, Burgers 2D/3D)?
   Are there observed failure cases or stability issues?

3. **Efficiency and resource usage:**
   What is the total training time and GPU memory compared to the baselines?
   Does the operator training overhead offset the convergence benefits provided by the curriculum?

4. **Operator bootstrapping bias:**
   Could early PINN errors accumulate within the operator during self-training?
   Beyond masking, have you tried uncertainty-weighted losses or consistency regularization between PDE residuals and operator outputs?

5. **Stronger baselines and ablations:**
   Please include comparisons with PINO/DeepONet + physics constraints, and report how the mask ratio and FNO resolution jointly affect performance and stability.

---

### Note · Authors · 2025-11-18

I have read and agree with the venue's withdrawal policy on behalf of myself and my co-authors.